# Dynamic redox and nutrient cycling response to climate forcing in the Mesoproterozoic ocean

Yafang Song [1] ✉, Fred T. Bowyer [1,2], Benjamin J. W. Mills [1], Andrew S. Merdith [1], Paul B. Wignall[1], Jeff Peakall[1], Shuichang Zhang[3], Xiaomei Wang[3], Huajian Wang[3], Donald E. Canfield[4], Graham A. Shields [5] & Simon W. Poulton [1]

Controls on Mesoproterozoic ocean redox heterogeneity, and links to nutrient cycling and oxygenation feedbacks, remain poorly resolved. Here, we report ocean redox and phosphorus cycling across two high-resolution sections from the ~1.4 Ga Xiamaling Formation, North China Craton. In the lower section, fluctuations in trade wind intensity regulated the spatial extent of a ferruginous oxygen minimum zone, promoting phosphorus drawdown and persistent oligotrophic conditions. In the upper section, high but variable continental chemical weathering rates led to periodic fluctuations between highly and weakly euxinic conditions, promoting phosphorus recycling and persistent eutrophication. Biogeochemical modeling demonstrates how changes in geographical location relative to global atmospheric circulation cells could have driven these temporal changes in regional ocean biogeochemistry. Our approach suggests that much of the ocean redox heterogeneity apparent in the Mesoproterozoic record can be explained by climate forcing at individual locations, rather than specific events or step-changes in global oceanic redox conditions.

Our understanding of the chemical evolution of the Precambrian ocean has evolved markedly over recent years. For the Mesoproterozoic (1.6–1.0 billion years ago; Ga) in particular, a stratified ocean redox model has emerged, with oxic surface waters commonly being underlain by euxinic (sulfidic) mid-depth waters in productive areas, with deeper waters dominantly being ferruginous (anoxic and iron-bearing)[1–3]. Emerging geochemical evidence suggests that Mesoproterozoic ocean redox chemistry was, however, considerably more variable than this generalized model implies. For example, while long-term paleo-records in some areas document dominantly ferruginous conditions with oxygenation of only very shallow waters[4], other studies suggest either a deepening of the oxycline[5–7], or the development

of dysoxic[8,9] or oxic deeper waters[10,11]. In addition, transitions between ferruginous and euxinic conditions have been documented within the same stratigraphic succession on multi-million-year timescales[12].

The factors that governed this variability in Mesoproterozoic ocean redox chemistry are essentially unknown. These uncertainties are at least partly due to inherent difficulty in distinguishing temporal from spatial variability when comparing limited, and often poorly-dated, Mesoproterozoic records from different settings and time periods. This also limits understanding of potential oxygenation feedbacks related to the availability of essential nutrients such as phosphorus, which was strongly impacted by variability in the precise redox state of the Mesoproterozoic water column[13–16].

[1]School of Earth and Environment, University of Leeds, Leeds LS2 9JT, UK. [2]School of GeoSciences, University of Edinburgh, James Hutton Road, Edinburgh EH9 3FE, UK. [3]Key Laboratory of Petroleum Geochemistry, Research Institute of Petroleum Exploration and Development, China National Petroleum Corporation, Beijing 100083, China. [4]Nordcee, Department of Biology, University of Southern Denmark, Odense 5230, Denmark. [5]Department of Earth Sciences, University College London, London WC1E 6BT, UK. ✉e-mail: eeyso@leeds.ac.uk

One factor that has largely been overlooked in terms of its impact on ocean redox chemistry and nutrient cycling in the Mesoproterozoic is the role of climate, both on longer (e.g., millions of years) and shorter (e.g., orbital) timescales, and in terms of the climatic regime at the paleogeographic location of individual study sites. Recently, a climatic control on orbital and longer (tens of millions of years) timescales has been documented in relation to the geochemistry of marine sediments from the ~1.4 Ga Xiamaling Formation on the North China Craton (NCC)[12,17]. However, detailed links between changes in climate, oceanic redox conditions and nutrient cycling have not specifically been explored, which limits understanding of controls on regional-scale ocean redox heterogeneity and hence links to global-scale oxygenation dynamics.

Here, we focus on two sections sampled at cm-scale, which comprise fine-grained siliciclastic rocks from the Mesoproterozoic Xiamaling Formation. We combine multiple redox proxies (Fe speciation, redox-sensitive trace element systematics, pyrite S isotopes) with elemental weathering indicators (K/Al, Ti/Al), and we utilize a phase partitioning approach to reconstruct P cycling. Our focus on two distinct high-resolution sections allows a detailed evaluation of climatic controls on ocean redox and nutrient cycling, both on orbital timescales (within sections) and over millions of years (between sections). We then utilize a biogeochemical multi-box model to explore how distinct regional climatic regimes may have instigated local variability in ocean biogeochemistry, thus providing a compelling explanation for the extensive ocean redox heterogeneity observed in the Mesoproterozoic paleo-record.

## Results and Discussion
### Geological setting
The investigated Xiamaling Formation succession was deposited in the Xiahuayuan region, NCC (Fig. 1)[12,17]. The Xiamaling Formation is commonly divided into six units, with the first four units (note that these units are numbered from the top of the section downwards) comprising fine-grained siliciclastic rocks that represent deep-water deposition near or below storm wave base[12,17]. High-precision zircon dating of a tuff layer in unit 2 and a bentonite layer at the top of unit 3 gave ages of $1384.4 \pm 1.4$ million years ago (Ma) and $1392.2 \pm 1.0$ Ma, respectively, which constrain deposition of the Xiamaling Formation to within the Ectasian Period of the Mesoproterozoic Era[17]. The Xiamaling Formation rocks in the study area are of very low thermal maturity, with burial temperatures dominantly below 90°C[17]. Paleogeographic reconstructions place the NCC at low latitudes (~15°N) during deposition of the Xiamaling Formation[18,19]. Further details of the geological setting are provided in the Supplementary Information.

### Redox geochemistry
Oceanic redox conditions were reconstructed for continuous sections from unit 1 (Section A) and unit 4 (Section B) of the Xiamaling Formation (Fig. 1; Fig. S1; see Methods for analytical procedures and Supplementary Information for details of the geochemical proxies used). In Section B, total organic carbon (TOC) concentrations are very low (~0.06–0.09 wt%), with slight peaks occurring in the green mudstone intervals (Fig. 2B). Highly reactive iron over total iron ($Fe_{HR}/Fe_T$) ratios show clear cyclicity through this section, with values below 0.22 in the green mudstone intervals, and above 0.38 in the red mudstone intervals (Fig. 2B). However, uranium enrichment factors ($U_{EF}$; see Methods) show the opposite trend, with mild enrichments (i.e., >1) in the green mudstone intervals, and values below 1 in the red mudstone intervals. Very low $Fe_{py}/Fe_{HR}$ ratios demonstrate negligible pyritization of the $Fe_{HR}$ pool throughout Section B (Fig. 2B), and this is accompanied by molybdenum enrichment factors ($Mo_{EF}$) that are also very low (≤2.0; Supplementary Data 1). Rhenium enrichment factors ($Re_{EF}$) are elevated throughout the section, while manganese enrichment factors ($Mn_{EF}$) are persistently low (Fig. 2B).

The geochemistry of Section A contrasts greatly with that of Section B. In Section A, TOC concentrations range from 2.43 wt% to 4.96 wt%, with clear cyclicity between relatively TOC-rich and TOC-poor intervals (Fig. 2A; Supplementary Data 1). All samples in this section have $Fe_{HR}/Fe_T$ ratios above 0.38, with $U_{EF}$ values that are consistently elevated (i.e., >1). In almost all cases, $Fe_{py}/Fe_{HR}$ ratios are above 0.6, while $Mo_{EF}$ values are also high, but with distinct cyclicity that broadly matches the TOC cyclicity (Fig. 2A; note that some geochemical profiles exhibit slightly offset responses, which we explore further in the Supplementary Information). Pyrite $\delta^{34}S$ data also show general cyclicity, whereby higher values tend to occur in the intervals where TOC concentrations and $Mo_{EF}$ values are lower (Fig. 2A).

### Nutrient and weathering proxies
In Section B, total P concentrations (expressed in terms of P/Al in ppm/wt%) are lower than the average upper continental crust (UCC) value (P/Al = 87 ppm/wt%)[20], with distinct peaks in the upper parts of the green mudstone horizons (Fig. 2B). Reactive P concentrations (P phases that are potentially bioavailable, expressed as $P_{reac}/Al$; see Methods) show similar trends to P/Al ratios, suggesting that reactive P exerts the dominant control on variability in the P/Al profile. Molar TOC/organic P ($P_{org}$) and TOC/$P_{reac}$ ratios are both considerably lower than the Redfield ratio (TOC/$P_{org}$ ratio = 106:1), with no apparent trends through the section (Fig. 2B). K/Al ratios are consistently higher than the UCC average (0.3), with no clear trends through the section, while Ti/Al ratios are also consistently above the UCC average (0.05), with slight peaks in the middle of green mudstone intervals, and lower values dominating in red mudstone intervals (Fig. 2B).

P/Al ratios are above the UCC average throughout Section A, but distinct troughs occur in the low TOC intervals, and these intervals are also characterized by similar troughs in $P_{reac}/Al$ ratios (Fig. 2A). Compared to the Redfield ratio, TOC/$P_{org}$ ratios in Section A are highly elevated, with distinct cyclicity that matches trends in TOC (Fig. 2A). TOC/$P_{reac}$ ratios are lower than TOC/$P_{org}$ ratios and show opposite cyclicity, but all values remain above the Redfield ratio (Fig. 2A). K/Al ratios are consistently above the UCC average and reach peaks in the intervals characterized by lower TOC, followed by a general progressive decrease to lower values (Fig. 2A). Ti/Al ratios are below the UCC average with no distinct change through the section.

### Redox cyclicity
For Section B, we reconcile the contrasting redox signals from $Fe_{HR}/Fe_T$ ratios and $U_{EF}$ values (Fig. 2B) by additionally considering the phase partitioning of the $Fe_{HR}$ pool, which is dominated by oxic phases (e.g., Fe (oxyhydr)oxides) in the red mudstone intervals, and reduced $Fe_{HR}$ phases (e.g., siderite) in the green mudstone intervals (Supplementary Data 1). Together, these geochemical signals suggest that the green mudstones were deposited under anoxic bottom water conditions (giving elevated $U_{EF}$ values)[21,22], but extensive reductive dissolution of Fe (oxyhydr)oxide minerals during early diagenesis promoted recycling of a proportion of the dissolved $Fe^{2+}$ to the water column (giving lower $Fe_{HR}/Fe_T$ ratios)[3,12,23]. This would promote ferruginous, rather than euxinic, water column anoxia, which is supported by very low $Fe_{py}/Fe_{HR}$ ratios (Fig. 2B) and low $Mo_{EF}$ values (Supplementary Data 1).

By contrast, the geochemistry of the red mudstones suggests deposition under less reducing conditions, where $Fe^{2+}$ was oxidized in the water column and precipitated as Fe (oxyhydr)oxide minerals. An oxygen minimum zone (OMZ) setting has previously been invoked for this part of the Xiamaling Formation[12]. Our high-resolution data support this scenario, with the green mudstones representing deposition within the ferruginous OMZ, and the red mudstones representing deposition beneath the OMZ during regular repeated intervals when the spatial extent of the OMZ contracted (Fig. 3; see discussion in the Supplementary Information). A fully oxygenated deeper ocean (i.e., below the OMZ) has previously been invoked[11,12]. However, elevated

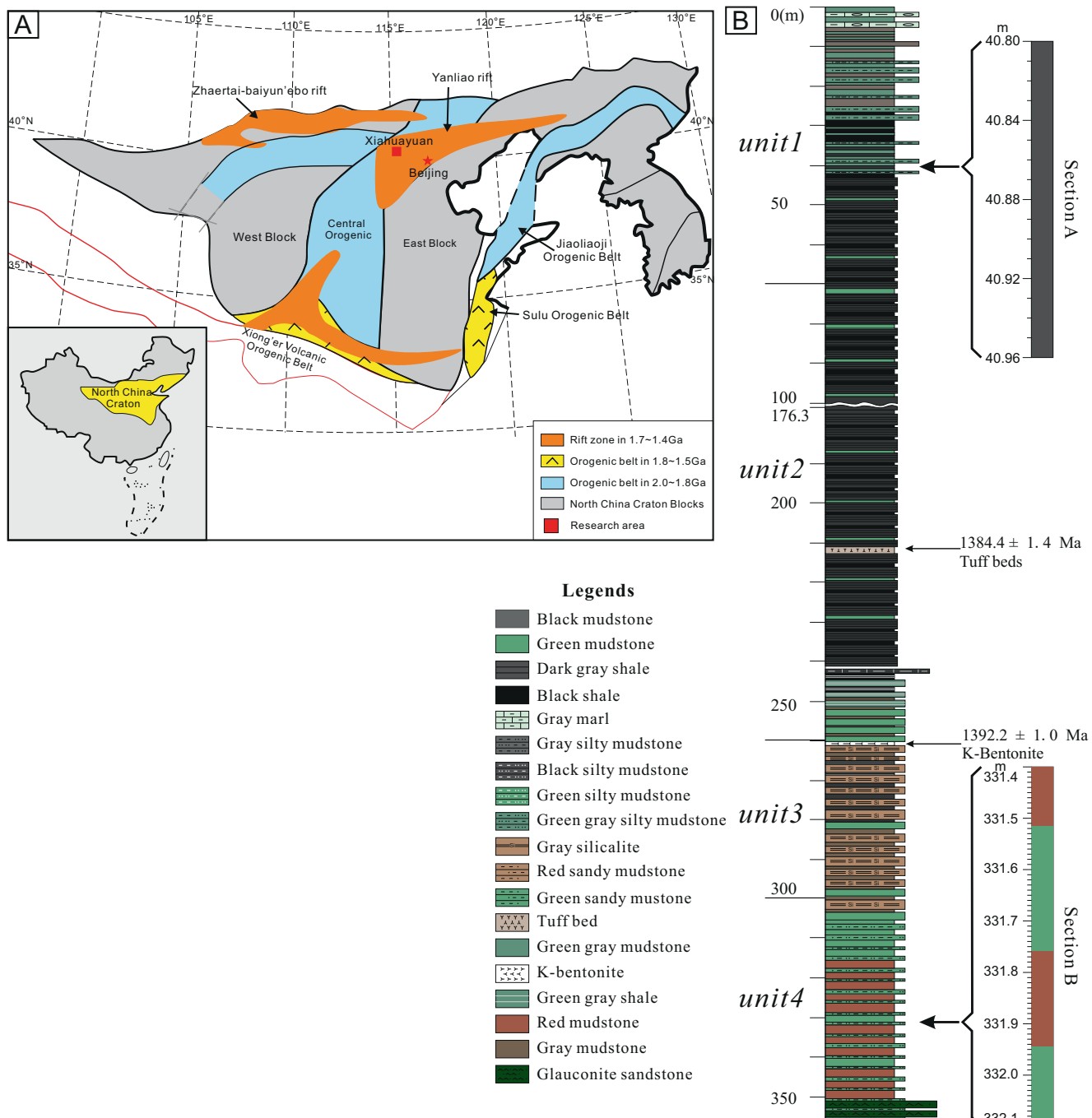

**Fig. 1 | Simplified geological map and stratigraphy of the Xiamaling Formation. A** Geological map of the North China Craton (NCC). Red square shows the study region. **B** Generalized stratigraphy of the 4 units of the Xiamaling Formation, indicating the positions of the two sampled high-resolution sections. Modified after ref. 12 with permission.

Re$_{EF}$ and depleted Mn$_{EF}$ values occur throughout the section (Fig. 2B). Observations from modern marine sediments[24,25] demonstrate that Re is enriched under intermediate redox conditions, where the oxygen penetration depth below the sediment-water interface is shallow (~1 cm). Such conditions may occur in sediments deposited beneath an oxic water column if sufficient TOC is available to promote oxygen depletion close to the sediment-water interface[26]. However, our red mudstone samples are characterized by particularly low TOC concentrations (0.070 ± 0.006 wt%), and hence elevated Re concentrations in the absence of elevated U (Fig. 2B) would be consistent with dysoxic bottom waters[24,25]. This is supported by very low Mn$_{EF}$ values (Fig. 2B), which suggests mobilization of Mn under oxygen-depleted conditions, in contrast to the Mn retention that might be expected in well-oxygenated settings (note that oxidation of Mn(II) to Mn(IV) requires a higher redox potential compared to Fe(II) oxidation[27]).

In Section A, elevated Fe$_{HR}$/Fe$_{T}$ ratios and U$_{EF}$ values are entirely consistent with persistent deposition beneath anoxic bottom waters[3,21–23]. Elevated Fe$_{py}$/Fe$_{HR}$ ratios (>0.6) suggest possible euxinia[28], and a sulfidic water column is supported by persistently elevated Mo$_{EF}$ values, requiring significant sulfide availability to convert the molybdate anion to particle-reactive thiomolybdate[29–31]. However, Fe$_{py}$/Fe$_{HR}$ ratios are at the lower end of the range typically taken to indicate euxinia (Fig. 2A), and Mo concentrations are relatively low compared to modern euxinic basins[32]. These results may relate to relatively low

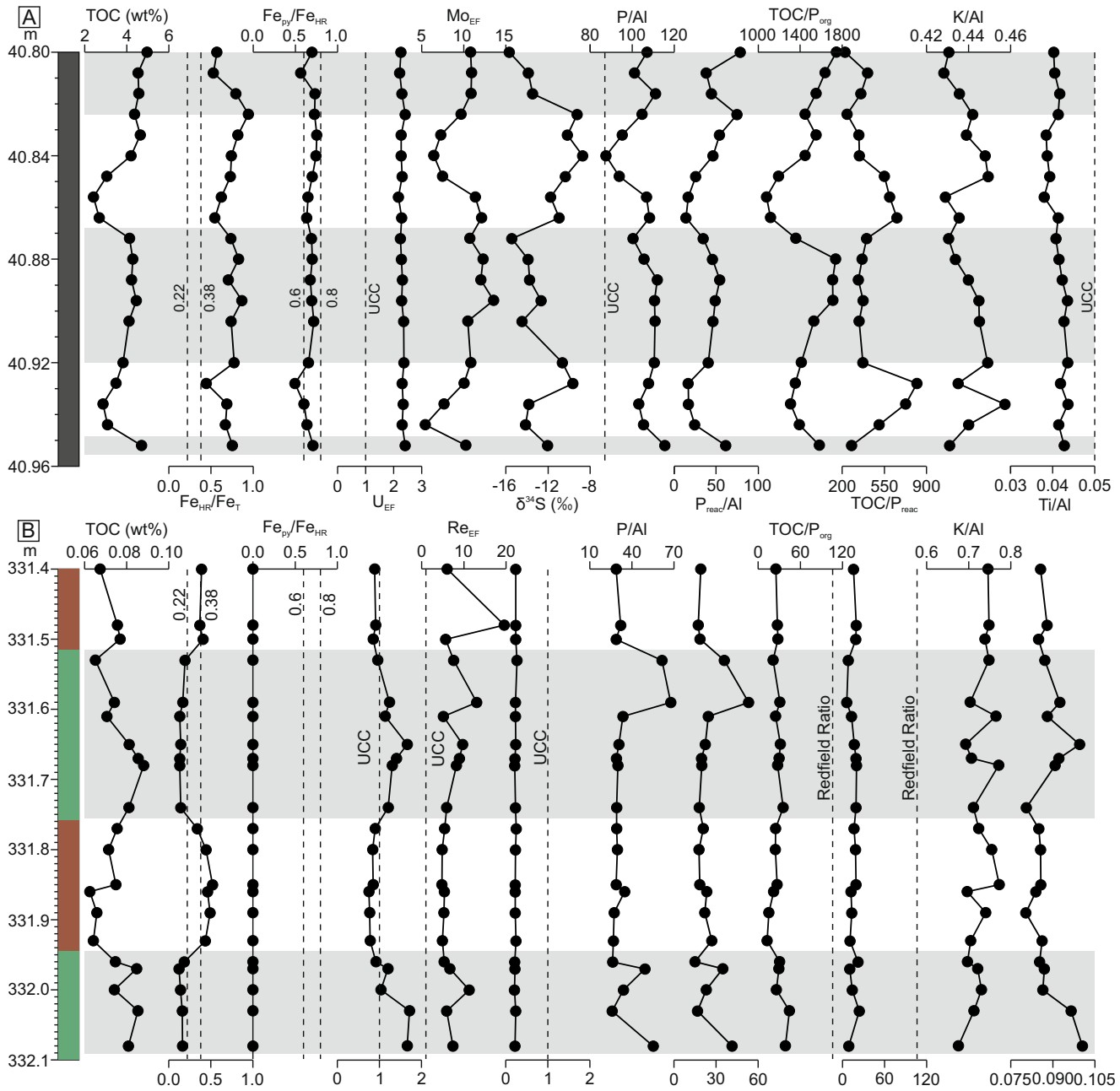

**Fig. 2 | Geochemical and isotopic profiles.** Section **A**, where shading indicates approximate position of cycles between lower total organic carbon (TOC) with more weakly sulfidic conditions (white), and higher TOC with more highly sulfidic conditions (gray). Section **B**, where shading corresponds to lithologic changes between red and green mudstones. Dashed lines on $Fe_{HR}/Fe_T$ profiles represent calibrated thresholds indicative of oxic (<0.22) and anoxic (>0.38) water column conditions[3], while dashed lines on $Fe_{py}/Fe_{HR}$ profiles represent the lower (0.6) and upper (0.8) thresholds for recognition of euxinia[28]. Dashed lines on elemental ratio profiles represent average Upper Continental Crust (UCC; P/Al = 87, Ti/Al = 0.05)[20]. Note that the K/Al ratio for UCC is 0.3, and thus falls below the scale shown on these plots. Dashed lines on $TOC/P_{org}$ and $TOC/P_{reac}$ profiles represent the molar Redfield ratio of 106:1. Elemental ratios are reported as wt%/wt%, except for P/Al and $P_{reac}$/Al, which are reported as ppm/wt%.

sulfide availability compared to modern euxinic basins, which would be an expectation given the low sulfate concentrations proposed for the Mesoproterozoic ocean[33]. Indeed, while oceanic Mo concentrations would also have been impacted by widespread Mo sequestration due to expansive ocean anoxia[34], the distinct cyclicity in $Mo_{EF}$ values (Fig. 2A) suggests that the redox state of the water column likely varied between a weakly and more strongly sulfidic state (see Supplementary Information for more detailed discussion). Since TOC concentrations are relatively high throughout the section (and thus microbial sulfate

reduction would not have been limited by TOC availability), the cyclicity between weakly and more strongly sulfidic conditions likely relates to changes in sulfate availability. This is supported by pyrite $\delta^{34}S$ data, whereby higher values tend to occur during the weakly euxinic intervals, which implies either enhanced limitation of sulfate reduction rates[12,35], or a relatively increased proportion of the pyrite forming under closed system conditions in the sediment pile[36], both of which are consistent with lower sulfate concentrations in the water column (see Supplementary Information).

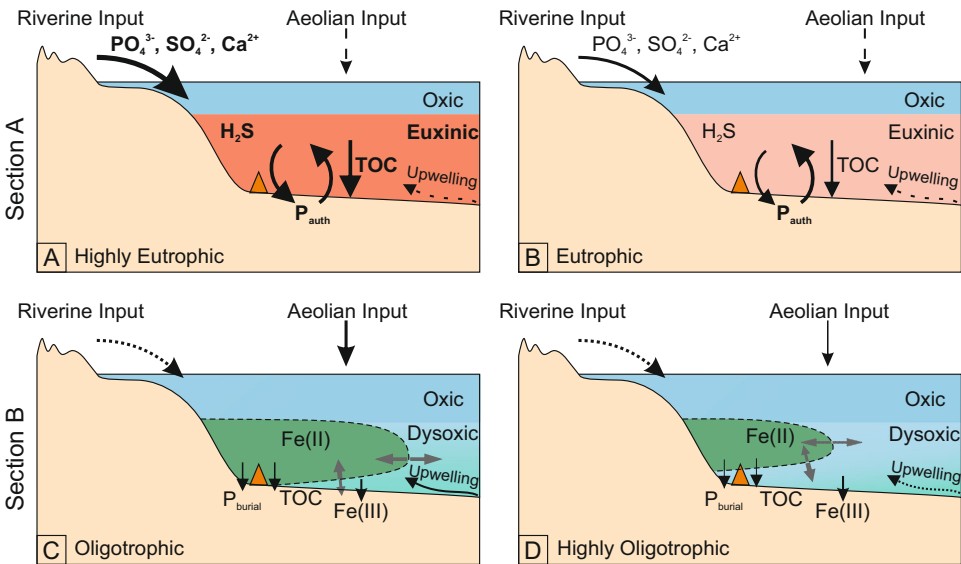

**Fig. 3 | Schematic diagram highlighting fluctuating C-S-Fe-P systematics during deposition of eutrophic Section A and oligotrophic Section B. A** Highly euxinic intervals in Section A experienced more intense runoff and organic matter burial. **B** Weakly euxinic intervals in Section A were driven by a relatively decreased riverine influx of sulfate and phosphate. **C** Deposition of Section B occurred within a ferruginous oxygen minimum zone (OMZ), with relatively increased aeolian inputs and upwelling under arid conditions promoting an expansion of the OMZ. **D** Deposition of Section B occurred beneath the ferruginous OMZ, where iron (oxyhydr)oxide minerals deposited under dysoxic bottom waters. A lower aeolian influx and upwelling limited the spatial development of the OMZ. Relative size of arrows indicates different flux intensities. Orange triangle indicates approximate depositional position for each scenario. Dashed lines indicate lower and/or more uncertain fluxes. TOC: total organic carbon.

## Nutrient cycling dynamics

Phosphorus bioavailability likely exerted a major control on primary productivity during the Mesoproterozoic Era, which in turn may have influenced longer- and shorter-term oxygenation dynamics via a variety of redox-linked feedbacks[13–16]. In Section B, $TOC/P_{org}$ and $TOC/P_{reac}$ ratios are well below the Redfield ratio (Fig. 2B). Such ratios can arise due to extensive oxidation of a limited supply of organic matter and the increased ability of bacteria to store P under better oxygenated conditions (giving low $TOC/P_{org}$ ratios)[37], coupled with additional P drawdown in association with Fe minerals (giving lower $TOC/P_{reac}$ ratios than $TOC/P_{org}$)[15]. In the case of the green mudstone intervals, the low $TOC/P_{reac}$ ratios suggest that Fe-bound P (along with organic P) was largely retained in the sediment (following "sink-switching" to authigenic carbonate fluorapatite; Supplementary Data 2), while $Fe^{2+}$ was more extensively recycled back to the water column as discussed above. During deposition of the red mudstones, the less reducing conditions also promoted retention of P in the sediment, dominantly as either authigenic carbonate fluorapatite or organic P, with only a minor post-diagenetic contribution from Fe-bound P (Supplementary Data 2). Ultimately, this limited degree of recycling, coupled with the potential for initial drawdown of P in association with Fe minerals precipitating in the water column, would have helped promote persistent oligotrophic conditions.

In Section A, molar $TOC/P_{org}$ ratios (Fig. 2A) are above both the Redfield ratio and the predicted ratio of ~300:1 for organic matter buried under the P-limited conditions sometimes inferred for the Mesoproterozoic[13]. The elevated $TOC/P_{org}$ ratios demonstrate extensive preferential release of P during anaerobic organic-matter remineralization, which is commonly particularly enhanced during microbial sulfate reduction[37]. Indeed, consistent with our suggestion of cyclicity in terms of sulfide availability (as constrained by cyclicity in $Mo_{EF}$ and pyrite $\delta^{34}S$ values), $TOC/P_{org}$ ratios are distinctly lower under weakly sulfidic conditions, relative to more highly sulfidic intervals (Fig. 2A).

Phosphorus that is recycled back to the water column, following organic matter remineralization and the reductive dissolution of Fe

(oxyhydr)oxide minerals, may promote a positive productivity feedback[38]. However, the extent of this recycling depends on the degree to which the P released during diagenesis is fixed as authigenic phases in the sediment[38–40]. As with Section B, $TOC/P_{reac}$ ratios are lower than $TOC/P_{org}$ ratios throughout Section A (Fig. 2), demonstrating 'sink-switching' and/or additional drawdown of water column P in association with other phases, such as Fe (oxyhydr)oxides.

Recycling of P back to the water column is, however, apparent throughout Section A, as indicated by $TOC/P_{reac}$ ratios that are above the Redfield ratio. However, despite a lower extent of P release from organic matter in weakly euxinic intervals (indicated by lower $TOC/P_{org}$ ratios; Fig. 2A), a higher proportion of the recycled P was mobilized back to the water column (indicated by higher $TOC/P_{reac}$ ratios) relative to highly sulfidic intervals. Nevertheless, because more TOC was deposited during highly sulfidic intervals, a similar flux of P was recycled under both weakly and highly euxinic conditions (assuming a relatively constant sedimentation rate; see Supplementary Information). Despite this P recycling, sediments in Section A are characterized by persistently elevated P/Al ratios and consistently high TOC concentrations (Fig. 2A), which suggests an overall productive setting throughout deposition of Section A. However, TOC and $P_{reac}$ ($P_{reac}$/Al profile; Fig. 2A) concentrations are higher during the highly euxinic intervals, despite the relatively constant recycling flux of P throughout Section A. This implies enhanced bioavailability of phosphate and more productive conditions during the more intensely euxinic intervals, which we explore below in terms of potential variability in the weathering influx of bioavailable phosphate.

## Orbital forcing of climate change

The short-term redox and nutrient cyclicity we document is consistent with the suggestion of an orbital climate forcing control during deposition of the Xiamaling Formation[17]. However, given the inherent difficulties in estimating the dominant orbital timescales for these cycles, our main aim here is rather to evaluate particular environmental controls on the observed cyclicity. To this aim, we evaluate potential weathering controls on redox-nutrient dynamics by

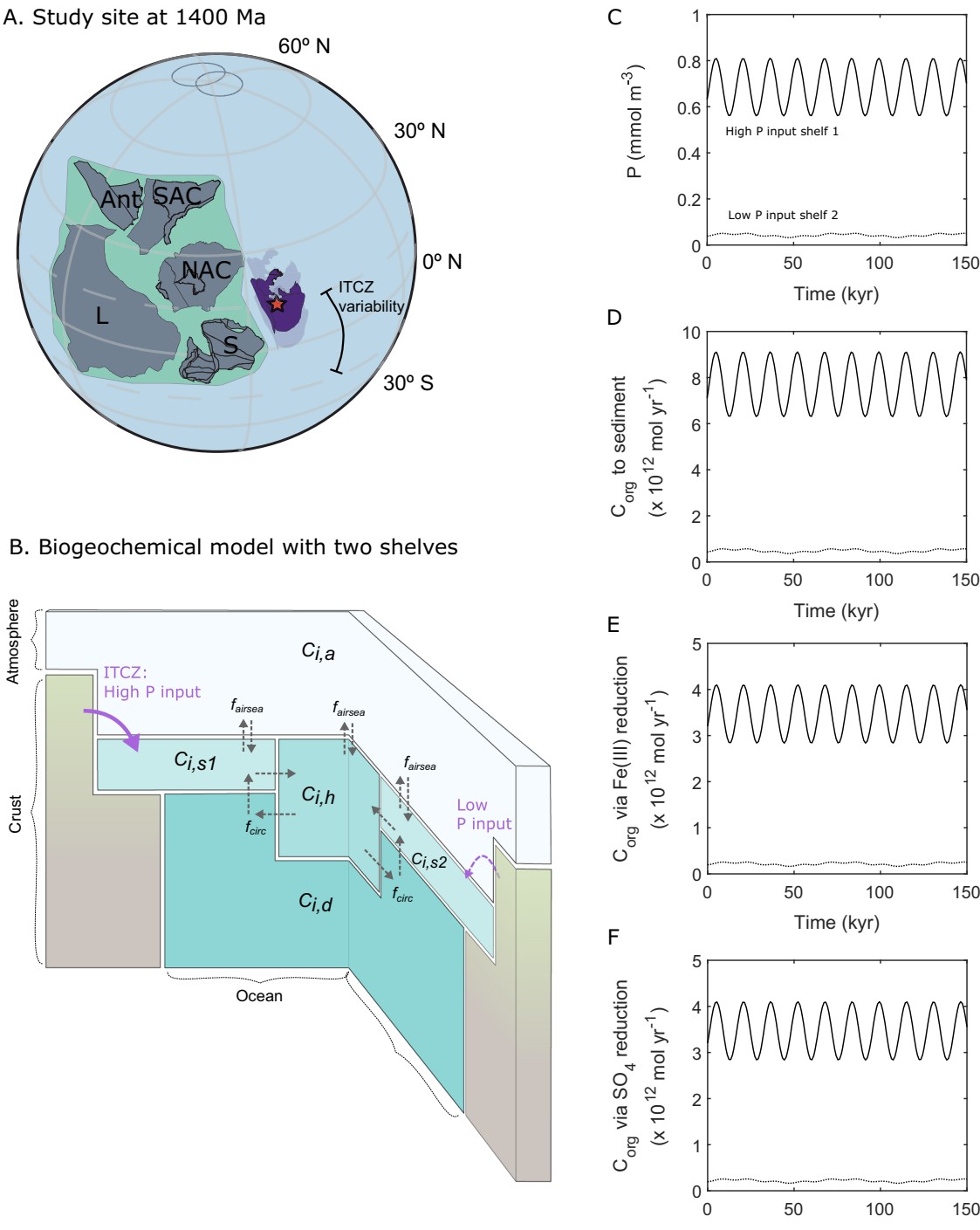

**Fig. 4 | Conceptional framework and outputs of the multi-box biogeochemical model. A** Paleogeographic reconstruction at -1.4 Ga, with red star indicating studied area. ITCZ: intertropical convergence zone. The ITCZ generally falls within -15° of the equator in both hemispheres[59, 60]. L, Laurentia; Ant, Antarctica; SAC, South Australian Craton; NAC, North Australian Craton; S, Siberia. Purple polygon represents the NCC, with transparent outlines representing the approximate uncertainty from the (temporally) surrounding palaeomagnetic data. Green polygon approximates interconnected landmasses during the tenure of the supercontinent Nuna[61–63]. Longitude is unconstrained, and the two circles on the north pole represent palaeomagnetic data from Laurentia that constrain Nuna to a low-latitude position at this time[64]. **B** Conceptional framework for the multi-box biogeochemical model with two shelves (Shelf 1 is shown on the left, with a solid arrow indicating a higher input of P, while Shelf 2 is located on the right, with the dashed arrow indicating a lower input of P). Biogeochemical cycling in different boxes responds to various weathering inputs on the two shelves. **C** A high weathering flux on Shelf 1 is set to oscillate at periods of 15 kyr, while a low weathering flux on Shelf 2 is set to oscillate at periods of 60 kyr, which is based on the assumption of constant sedimentation rates (see Supplementary Information). However, given that a constant sedimentation rate was highly unlikely, these forcings are applied solely to demonstrate the range of potential fluctuations in ensuing biogeochemical characteristics, and the actual periodicity of these fluctuations may have varied considerably. **D** High and variable organic carbon burial rates ($C_{org}$) occur on Shelf 1, while low and relatively stable rates occur on Shelf 2. **E** High and variable $C_{org}$ consumption rates occur due to Fe(III) reduction on Shelf 1, with very low (but variable) remineralization rates by Fe(III) reduction on Shelf 2. **F** High and variable $C_{org}$ consumption rates occur due to sulfate reduction on Shelf 1, with very low remineralization rates by sulfate reduction on Shelf 2.

considering different elemental weathering proxies. In Section B, consistently high K/Al ratios suggest that chemical weathering rates in the sediment source region were likely low and relatively stable, due to the diminished solubility of K as weathering intensity decreases (Fig. 2B)[41,42]. However, high Ti/Al ratios, while also reflecting the primary chemical nature of the source material, imply a significant contribution of sediment from aeolian sources[43–45]. Indeed, since there is no apparent reason to assume large-scale, repetitive changes in the geographic area being weathered (i.e., the sediment source area) during deposition of Section B, peaks in Ti/Al around the mid-point of green mudstone intervals likely document maxima in the aeolian source flux. These combined observations imply a wind-driven orbital control in an arid environment[12]. We suggest that this control likely drove OMZ dynamics via a combination of changes in aeolian sediment fluxes and possible changes in the intensity of upwelling (and hence P supply).

In Section A, peaks in K/Al ratios in weakly euxinic intervals (Fig. 2A) are consistent with a lower intensity of chemical weathering, and gradual transitions to lower values in more highly sulfidic intervals suggest progressively more intense chemical weathering, due to the enhanced solubility of K as weathering intensity increases[41,42]. In addition, the low and relatively constant Ti/Al ratios (Fig. 2A) support a limited and stable wind-driven aeolian contribution[43–45], with a dominant weathering influx from continental runoff. The intervals of more intense chemical weathering would be expected to result in increased delivery of dissolved phosphate and sulfate in runoff (Fig. 3A), the latter of which is supported by the lower pyrite $\delta^{34}S$ values (Fig. 2A), despite more intense sulfide generation. In this overall eutrophic setting, this would have promoted enhanced productivity and TOC burial, along with the development of more intense euxinia, all of which would have been exacerbated by P recycling from the sediments (Fig. 3A). By contrast, periods of less intense chemical weathering would have resulted in a lower influx of both phosphate and sulfate. Whilst a lower sulfate influx would have promoted less intense euxinia, the overall flux of recycled P from sediments remained similar. Therefore, lower phosphate availability in surface waters, which occurred largely due to the lower weathering influx, would have led to relatively lower rates of productivity and TOC burial (Fig. 2A).

## Mesoproterozoic ocean redox heterogeneity driven by regional climate dynamics

Our ocean redox and nutrient data document a highly dynamic response to orbital climate forcing in the Mesoproterozoic Era. More broadly, since our samples originate from the deepest part of the Xiamaling basin with no clear change in water depth between the two sections[6] (and hence a sea-level control on the biogeochemical dynamics is highly unlikely), the distinct contrasts in ocean biogeochemistry between sections A and B are consistent with the suggestion of longer-term climate forcing driven by shifts in the position of the intertropical convergence zone (ITCZ) and related atmospheric circulation cells[12,17,46]. This climatic variability may also have been impacted by a possible southward shift of the NCC during the depositional period from ~1.44 Ga to 1.35 Ga[18,19]. Paleogeographic reconstruction places the study site at about 15˚N at 1.4 Ga (Fig. 4A; Supplementary Information), which according to climate modeling, has been the approximate position of the variable northern limit of the ITCZ, at least over Phanerozoic time[47]. Thus, the arid, aeolian conditions documented by our geochemical data for Section B indicate deposition outside the influence of the ITCZ, probably near the northern limb of an ancient Hadley Cell. By contrast, Section A appears to have been deposited in a region with a strong contribution from continental runoff, consistent with a low latitude, full tropical setting under direct influence of the ITCZ.

We explore this further by running a multi-box biogeochemical model that considers the mixing of water between the surface ocean,

deep ocean and a globally representative continental shelf, alongside changes in weathering inputs (Fig. 4B)[48]. The model is modified to include two different surface ocean boxes that overlie different continental shelves and have unique weathering inputs, to explore whether these ocean margins can maintain different carbon cycle dynamics simultaneously, depending on ITCZ influence (see Methods). Shelf 1 is subject to a high weathering influx and high upwelling rate, while Shelf 2 is subject to a much lower weathering influx and upwelling rate, both taken to be two orders of magnitude lower. We alter both the weathering and upwelling supply of P because in this relatively simple model, all deep-water recycling transits through the continental shelves and open-ocean upwelling is not considered. Thus, the model shelves are likely overly connected to the deep ocean phosphate supply which artificially limits the effects of changes in weathering rates. We allow weathering fluxes to oscillate at periods of 15 kyr and 60 kyr, respectively, consistent with our broad estimates for the duration of the orbital cycles (see Supplementary Information), but note that while this affects the frequency of the modeled cycles at each site, it does not affect the contrasting million-year-scale biogeochemical dynamics observed between the two study sections.

The model outputs (Fig. 4C–F) show that changes in phosphate input result in substantially different (about tenfold) phosphorus concentrations and unique cyclic variations in each area, despite their mutual connectivity to the deep ocean interior. This drives a similar degree of variability in the amount of organic carbon ($C_{org}$) reaching the sediment in each area. The remineralization pathways in the model are simplified. We assume a simple burial efficiency of 10%, and that iron reduction processes a fixed fraction of sedimentary $C_{org}$, with the remainder being remineralized via sulfate reduction. In the oligotrophic Shelf 2 setting, $C_{org}$ is less available and is significantly depleted via Fe reduction, and little sulfide is produced. Furthermore, Fe reduction rates fluctuate, which would affect the size of the ferruginous OMZ, as documented by our data (Fig. 3C, D).

In the eutrophic Shelf 1 setting, sulfate reduction rates are high and variable, which would be expected to lead to the observed oscillations between weakly and strongly euxinic waters (Fig. 3A, B). Although highly simplified, our model confirms that variations in weathering inputs should be able to drive substantially different carbon cycle and redox signatures on different parts of the continental shelf. We additionally explore the model outputs when upwelling rates are evenly split between boxes, and these also show high and variable sulfate reduction rates in Shelf 1 coexisting with low sulfate reduction rates in Shelf 2, although the strong connections to the deep ocean synchronize the timescales of these changes in this model version (Fig. S7). We thus hypothesize that variations in the ITCZ position over the period of deposition (Fig. 4A) resulted in the Xiamaling Formation recording first our "Shelf 2" signal (i.e., Section B) and then our "Shelf 1" signal (i.e., Section A), with both signals likely coexisting in multiple locations throughout the time period.

Our combined approach demonstrates how ocean redox and nutrient cycling varied as a function of location and specific positioning relative to the ITCZ and global atmospheric circulation cells, which drove large-scale variability in ocean redox conditions and associated nutrient cycling, and more subtle, regional-scale orbital timescale variability. Unlike more recent time periods, this influence of shorter- and longer-term climate forcing has rarely been factored into studies evaluating Mesoproterozoic ocean redox chemistry and nutrient cycling, which instead often assume that the biogeochemistry of one region reflects the global ocean, and that any apparent "heterogeneity" reflects an "event" or "transition" to a new redox state. Future studies should therefore directly consider the paleogeographic location of study sites and the potential role of climate forcing, when expanding regional observations to the global scale, which in turn will impact understanding of Mesoproterozoic oxygenation dynamics and its role in biological evolution.

## Methods

### Elemental analyses

TOC concentrations were analyzed on a LECO carbon analyzer, with TOC determined after inorganic carbon removal using 20% HCl. Replicate analyses of $0.717 \pm 0.027$ wt% and $10.80 \pm 0.26$ wt% carbon reference materials yielded relative standard deviations (RSDs) of 3% and 5%, respectively, with an accuracy of >95%. Major and trace element concentrations were measured by inductively coupled plasma–optical emission spectrometry (Fe, P, Al, K, Ti) and inductively coupled plasma–mass spectrometry (U, Mo, Re, Mn) following an $HNO_3$-HF-$HClO_4$-$H_3BO_3$ digest on ashed (550°C for 8 h) samples[49]. Precision and accuracy were monitored by analyzing international sediment standard SGR-1b, with a RSD of <5% for all elements except Mo, which had a RSD of 12%. Accuracy was within 10% for all elements. Enrichment factors for each element ($X_{EF}$) were calculated relative to average UCC[20,50], as:

$$X_{EF} = (X/Al)_{sample}/(X/Al)_{UCC} \qquad (1)$$

### Fe-S systematics

Sequential Fe extractions were performed following standard operationally defined protocols[28,51]. A sodium acetate solution at pH 4.5 for 48 h at 50 °C was used to target carbonate-associated Fe ($Fe_{carb}$); a room-temperature sodium dithionite solution (pH 4.8 for 2 h) was used to target Fe (oxyhydr)oxide minerals ($Fe_{ox}$); and a room-temperature ammonium oxalate (6 h) was used to target magnetite ($Fe_{mag}$). Iron concentrations in the extraction solutions were measured via atomic absorption spectrometry. Pyrite concentrations ($Fe_{py}$) were determined via gravimetric precipitation of $Ag_2S$ following a standard chromous chloride extraction, with samples first checked for acid volatile sulfide (not present) using boiling 6 N HCl[52]. Accuracy and precision were monitored relative to international Fe speciation standard, WHIT[49], with a RSD of <5% and accuracy >94% for all phases. Highly reactive Fe ($Fe_{HR}$) was calculated as the sum of $Fe_{carb}$, $Fe_{ox}$, $Fe_{mag}$ and $Fe_{py}$[53]. Further details on the application of Fe speciation to paleoredox studies are provided in the Supplementary Information.

Pyrite-S isotope ($\delta^{34}S$) analyses were performed on $Ag_2S$ precipitates (no $Ag_2S$ was recovered for Section B samples) using an Elementar PYRO cube coupled to an IsoPrime continuous flow mass spectrometer. Pyrite $\delta^{34}S$ values are reported relative to the international Vienna-Canyon Diablo Troilite. Laboratory standard SWS-3A ($BaSO_4$, $\delta^{34}S = +20.3‰$), inter-laboratory standard CP-1 (chalcopyrite, $\delta^{34}S = -4.56‰$), and international reference standard IAEA S-3 ($\delta^{34}S = -32.06‰$) were used to check precision and accuracy. Replicate analyses of a laboratory $BaSO_4$ standard yielded an accuracy of <0.33‰, and a precision of <0.3‰ (1 σ) for $\delta^{34}S$.

### Phosphorus phase partitioning

Phosphorus phase partitioning was performed with an established sequential extraction scheme[54] adapted for ancient sedimentary rocks (Table S1)[55]. Four operationally-defined phosphorus pools were targeted, including iron-bound P ($P_{Fe}$), authigenic carbonate fluorapatite-associated P ($P_{auth}$), detrital apatite ($P_{det}$), and organic-bound P ($P_{org}$). Phosphorus concentrations were determined spectrophotometrically using the molybdate-blue method on a Spectronic GENESYS™ 6 at 880 nm, except for steps I, IV and V (Table S1), where the reagents interfere with the molybdate complex. For these steps, P was measured by inductively coupled plasma–optical emission spectrometry. Replicate analyses gave RSDs of <5% for each step, apart from $P_{Fe2}$, where the RSD was 18% due to the low concentrations of P in this phase.

### Biogeochemical modeling

The model developed here is modified from a simple 4-box atmosphere-ocean carbon-alkalinity system[48], which is based on a combination of similar systems in the literature[56–58]. All changes made for the present study are documented in this section and the full model equations are presented in the Supplementary Information.

The model environment is expanded from a single surface box with a high latitude deep water cycle, to include two surface boxes, Shelf 1 and Shelf 2, while retaining the high latitude deep-water box. An upwelling fraction (*Ufrac*) determines the fraction of deep water that enters each box, and this was also used to set the other circulation parameters for steady state. A weathering fraction (*Wfrac*) controls how the globally-calculated weathering inputs are split between the two surface boxes. For our experiments, which test an end-member case, both *Ufrac* and *Wfrac* were set to 0.99 for Shelf 1 and 0.01 for Shelf 2, which means that Shelf 1 receives 99% of the global upwelling water flux and 99% of the global weathering flux, with Shelf 2 receiving only 1% of these. As noted in the main text, this 99:1 split in nutrient inputs drives a more muted ~9:1 split in the calculated nutrient inventory and carbon export, due to the interconnectedness of the model ocean. A smaller 9:1 split in P inputs drives only a ~2:1 ratio in the shelf nutrient inventories and carbon export. Carbonate burial in each surface box is driven by the saturation state in that box, while organic carbon burial is scaled by the phosphorus concentration in the box, with P being delivered through weathering and circulated as a model tracer and buried in association with organic matter.

To replicate the orbital forcing, a sinusoidal multiplier was applied to the weathering fluxes into Shelf 1 and Shelf 2. These were set to oscillate on approximately 15 kyr and 60 kyr timeframes, where $t$ is time in model years (upwelling fluxes are not modified in this way):

$$W_{15} = 1 + \sin\left(4 \cdot t \cdot 10^{-4}\right) \qquad (2)$$

$$W_{60} = 1 + \sin\left(t \cdot 10^{-4}\right) \qquad (3)$$

A simplified remineralization scheme was applied to the model sediment-water interface to demonstrate how altered carbon supply in each shelf could drive cyclicity in the severity of euxinic conditions or in the extent of a ferruginous OMZ. We first calculated the available organic carbon from the overall model $C_{org}$ burial flux at the present day ($k_{mocb}$), a relationship to local shelf P concentration relative to the present day, and a simple burial efficiency (*BE*) set to 10%.

$$C_{sed_i} = \frac{k_{mocb}}{2} \cdot \frac{P_{Si}}{P_{S0}} \cdot \frac{1}{BE} \qquad (4)$$

For our test case we assumed a constant fraction of the available carbon is processed through Fe(III) reduction and the remainder is processed via $SO_4$ reduction:

$$remin_{Fe_i} = 0.5 \cdot C_{sed_i} \cdot (1 - BE) \qquad (5)$$

$$remin_{S_i} = C_{sed_i} \cdot (1 - BE) \cdot (1 - remin_{Fei}) \qquad (6)$$

Organic carbon burial is calculated as:

$$mocb_{sed_i} = C_{sed_i} \cdot BE \qquad (7)$$

## Data availability

The original data generated in this study are provided in the Supplementary Data Files 1, 2.

## Code availability

The code for the multi-box biogeochemical model is freely available at github.com/bjwmills/CARMER.

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

## Acknowledgements

This work was funded by a China Scholarship Council-University of Leeds Scholarship to Y.S. NERC grant NE/R010129/1 to S.W.P., B.J.W.M., F.T.B. and G.A.S., and NERC grant NE/T008458/01 to S.W.P. and F.T.B.

## Author contributions

The study was conceived by S.W.P and G.A.S; S.Z., X.W., H.W., and D.E.C. provided access to samples; Y.S., F.T.B., and S.W.P. collected samples; Y.S. performed geochemical analyses; Y.S. and B.J.W.M. performed biogeochemical modeling; A.S.M. provided paleogeographic constraints; P.B.W., J.P., and D.E.C. provided sedimentological insight; Y.S. and S.W.P. wrote the original version of the manuscript, with contributions from all co-authors.

## Competing interests

The authors declare no competing interests.
