## [Peer Review File · Nature Communications]

REVIEWER COMMENTS

Reviewer #2 (Remarks to the Author):

Song and colleagues present extremely-high-resolution geochemical investigation on the Xiamaling Formation, and demonstrate the control of orbital climate forcing on oceanic redox and nutrient dynamics based on the redox and weathering proxies, phosphorus phase and biogeochemical modelling. The study is interesting, important, and valuable, and the manuscript is well organized and easy to read. I suggest publication after a moderate revision. My detailed comments are as follows.

Major comments:

1. In this study, the authors used redox-sensitive elements, Fe speciation, and S isotopes as tracers for oceanic redox condition, K/Al and Ti/Al ratios for chemical weathering intensity, and P speciation for P cycling. I suggest to provide more backgrounds of these proxies at least in Supplementary Information, especially for weathering proxies (K/Al and Ti/Al ratios).
2. In this study, authors used K/Al ratio to trace chemical weathering intensity, based on the enhanced solubility of K as an indicator of increased weathering intensity (Line 208–209), and thus higher K/Al ratios in sediments indicate higher intensity of chemical weathering. I think this application is on the basis that detrital clay minerals and rock debris in sediments are reliable archives for terrestrial inputs. However, all the K/Al ratios of the Xiamaling Formation reported in this study are higher than that of the UCC (~0.3), seemingly indicating that there was seawater-derived K in these siliciclastic rocks. Please clarify this clearly (also see Major comment 1).
3. Tables S5 and S6 only present raw geochemical data. As shown in Fig. 2 and discussed in main text, the calculated values of Fe_{HR}/Fe_T, Fe_{py}/Fe_{HR}, UEF, MoEF, ReEF, MnEF, P/Al, Preac/Al, TOC/Porg, TOC/Preac, K/Al, and Ti/Al are better to be presented in tables.

Minor comments:

Line 21–26. I would suggest to introduce the reconstruction result from the lower section first, to the upper section then.

Line 41. What does “similarly” mean?

Line 69. I suggest that at least briefly introduce the background information of the studied area, stratigraphy, and samples, before the geochemical results in the main text, although details have been presented in Supplementary Information.

Line 70–71. As the authors only investigate two short intervals of the Xiamaling Formation, 16 cm of Unit 1 (Section A) and 70 cm of Unit 4 (Section B), thus the words “through unit 1 (Section A) and unit 4 (Section B)” is not appropriate here.

Line 76. As enrichment factors are calculated relative to average UCC, thus “relative to upper continental crust (UCC)” is not needed and should be deleted here.

Line 80. “values” cannot be “heavier”.

Line 81. How about Re and Mn enrichment factors of Section A?

Line 88. Fepy/FeHR ratios and MoEF values of Section B are extremely low, which should be also mentioned here.

Line 106. Add “those” between “than” and “in”.

Line 112. “ratios” and “values” cannot be “enrichments”.

Line 119. You just stated that a sulfidic water column is supported by persistently elevated MoEF values requiring significant sulfide availability in Line 115-116, which seems inconsistent with the sentence here. Please clarify this inconsistency.

Line 126. “higher” rather than “heavier”.

Line 128. It is better to present the contrasting reconstruction results of FeHR/FeT ratios and UEF values first.

Line 131–134. I think the high TOC contents in the green mudstone intervals can also be an important piece of evidence for anoxic bottom water conditions.

Line 143–144. A fully oxygenated deeper ocean was proposed based on the geochemical evidence of the Unit 3 (Zhang et al., 2016).

Line 160–161. The P/Al ratios of Section A are not low, inconsistent with the supposed P-limited conditions.

Line 164. $\delta^{34}\text{S}$ values can also be included.

Line 171. (1) “lower than” rather than “below”. (2) the Preac includes Porg, PFe, and Pauth, and thus TOC/Preac ratios should be usually lower than TOC/Porg ratios. Why not directly present the PFe and Pauth enrichments instead?

Line 180–182. Reorganize this sentence.

Line 190. See the comments on Line 171.

Line 206. See Major comment 2.

Line 207. Add “of” between “intensity” and “chemical”.

Line 213. “lower” rather than “lighter”.

Line 216–219. Reorganize this sentence. The lower sulfate input would have promoted less intense euxinia, which ultimately lower the flux of recycled P from sediments.

Line 220. See Major comment 2.

Line 227. Cite Wang et al. (2017) here.

Line 228. Is there evidence for the changes in the upwelling intensity?

Line 232–233. How do you know / What is the evidence, for the two sections in this study being “the deepest part of the Xiamaling basin”?

Line 238. Add “Ga” between “1.44” and “to”

Line 240–244. Unit 4 of Section B was deposited before Unit 1 of Section A, so I suggest to reorganize two sentence.

Line 267. “the mixed layer” of what, chemocline?

Line 273. Need to change to “approaches demonstrate”.

Line 273–276. Merge two sentences.

Line 276. “shorter-”.

Line 315–316. Move this sentence before the sentence of “Laboratory standard SWS-3A (BaSO_4 , $\delta^{34}\text{S} = +20.3\text{‰}$), inter-laboratory standard CP-1 (chalcopyrite, $\delta^{34}\text{S} = -4.56\text{‰}$), and international reference standard IAEA S-3 ($\delta^{34}\text{S} = -32.06\text{‰}$) were used to check precision and accuracy”, since you had reported $\delta^{34}\text{S}$ values before.

Line 326. “RSDs”.

Line 533. Remove “ $\text{K}/\text{Al} = 0.3$ ”.

Figure 2. (1) I suggest to make items of the geochemical profiles of Sections A and B consistent, namely, add ReEF and MnEF profiles for Section A and add MoEF profile for Section B; (2) the line representing the K/Al ratio of the UCC has not been shown and needs to be added.

Figure 4. Marker Shelf 1 and 2 in (B).

Comments on Supplementary Information:

Line 601. “High precision”.

Line 607–608. The Xiamaling Formation is divided into four members, Member 1 to 4 for bottom to top, or into six units, Unit 1 to 6 from top to bottom.

Line 610. “siliciclastic rocks”.

Line 612–614. The Xiamaling Formation is more than 400 m in thickness. You should explain why you only focused on two short intervals of Unit 1 (Section A) and Unit 2 (Section B).

Figure S1. Please add plotting scales for Sections A and B.

Reviewer #3 (Remarks to the Author):

General comments:

Song et al. present a variety of high resolution data exploring the influence of climate fluctuations on geochemical cycling within a Mesoproterozoic environment. Data include a variety of analyses (Fe spec, P spec, S isotopes, major and trace elements, and TOC) and are of good quality. Interpretations of redox state are very solid; authors do an excellent job of integrating findings from each analysis into a coherent geochemical story. The authors propose orbital scale climate variation in the form of ITCZ and Hadley cell changes as drivers of redox and nutrient fluctuation in their study environment. These appear to be reasonable hypotheses. The article's supplementary information is very useful and goes into great detail about aspects of the paper. Additionally, the manuscript is very well written and pleasant to read.

We did notice that the methods, results, and some interpretations in this manuscript are similar to those in Wang et al. 2017. We recognize that this manuscript brings new insights, so we just recommend that it be made explicit whether or not any of your data derive from Wang et al. 2017 and how your paper builds upon this previous study.

All in all, we believe this manuscript is of high quality and worthy of publication. We propose some minor changes. In a few places, we suggest the authors should qualify some of their hypotheses and/or acknowledge alternative scenarios. We also suggest they include some additional information in places, but nothing critical to the paper's main points.

Sincerely,

Amy Hagen and Benjamin Gill

Specific comments (also included in attached Word doc):

Line 118-123: We are not sure you can differentiate their proposed scenario that invokes local changes in sulfide availability driving Mo enrichments from one where the enhanced weathering also just delivers more Mo to the basin (with the sulfate). Both would lead to intervals with higher Fe(pyrite)/FeHR ratios, lower $\delta^{34}\text{S}$ of the pyrite and higher Mo enrichments. We would recommend the authors also discuss this possibility; it doesn't greatly change their interpretations.

Line 126-127: It might be nice to show a cross plot or stratigraphic plot with only these two sets of data in the supplement to more clearly illustrate this point.

Line 129: It might be best to qualify what is meant by "mineralogy" here, since Fe spec targets operationally defined pools and therefore we don't know specific minerals.

Line 131-134: Given the higher Fe/Al ratios (of the green intervals versus those in Section A), it might be worth mentioning the potential for clay to be a sink of highly reactive Fe.

Line 175-178: I'm not sure 'rate' is the correct word here. To compare rates we need to know the amount of time it took to produce the TOC/P ratio. 'Amount' might be a better word.

Line 179-180: This is probably not a realistic assumption. It's fine that it was assumed (it may be the best we can do), but we would like to see this statement better qualified.

Line 209-210: It would be better to cite the study/studies which ground truth Ti as a dust proxy rather than these two sources which simply apply the proxy.

Line 217-218: Shouldn't P recycling have declined as organic matter production declined?

Line 220-221: Low chemical weathering rates seem at odds with the elevated Fe/Al ratios for the green intervals in Section B compared to all of Section A. May suggest that either there is a difference in the weathering flux of iron (source or intensity) between these sections.

Line 224: The authors should be specific here. What large-scale changes are they referring to: overall weathering rates, type of material being weathered, and/or amount of geographic area being weathered?

Line 232-234: I'm not sure you can rule this out. Estimating paleowater depths is difficult, so I don't think you can simply say it is not a factor. Changes in terrigenous input could be factor.

Line 235-236: This seems to be a reasonable hypothesis, but are there potentially other climate influence factors? It may be worth mentioning some alternatives.

Line 250: It could be useful to mention why you've chosen to couple these two variables or what model outputs would look like if you changed just one of these factors.

Line 278-280: This seems like a bit of an overgeneralization, I'd suggest changing the wording a little bit here.

Line 329: I am not a box model expert so I can't offer much feedback here, but all looks good and seems to be explained well.

Line 521: I would suggest using something other than just a red arrow to indicate the studied sections. Perhaps a curly bracket would be helpful in showing that you're zooming in on a portion of the big column?

Line 572: I'd suggest adding a reference here so we know why you've shown that particular range for the ITCZ.

Line 678: Again, I'm not sure this is likely to be true. After all, you just discussed turbidites which represent episodic sedimentation. It could be good to just put in a short note acknowledging this.

Line 734-735: Since you mention this, it might be useful to elaborate a bit on why you don't see this being a problem in your section.

Line 738-739: This section might be a bit confusing for those who haven't read Pasquier et al. I'd suggest adding a bit about what they claim in the paper before you proceed to counter it. However, I also think that the vast majority of papers support Fe spec as a proxy and you may not need to dedicate so much space to addressing this. I think your last 2 sentences sum up your point nicely. This is just an opinion and not critical to your paper so feel free to keep this section as is if you like.

Line 742-746: As mentioned above, I think these two sentences emphasize an important point: that regardless of any doubts in Fe spec, your other data fully support your interpretations.

REVIEWER COMMENTS

Reviewer #2 (Remarks to the Author):

Song and colleagues present extremely-high-resolution geochemical investigation on the Xiamaling Formation, and demonstrate the control of orbital climate forcing on oceanic redox and nutrient dynamics based on the redox and weathering proxies, phosphorus phase and biogeochemical modelling. The study is interesting, important, and valuable, and the manuscript is well organized and easy to read. I suggest publication after a moderate revision. My detailed comments are as follows.

Response: We are grateful for the positive response and the useful comments that we hope we have been able to address in full.

Major comments:

1. In this study, the authors used redox-sensitive elements, Fe speciation, and S isotopes as tracers for oceanic redox condition, K/Al and Ti/Al ratios for chemical weathering intensity, and P speciation for P cycling. I suggest to provide more backgrounds of these proxies at least in Supplementary Information, especially for weathering proxies (K/Al and Ti/Al ratios).

Response: This is a good point, and we have provided more information of these proxies in the revised Supplementary Information.

2. In this study, authors used K/Al ratio to trace chemical weathering intensity, based on the enhanced solubility of K as an indicator of increased weathering intensity (Line 208–209), and thus higher K/Al ratios in sediments indicate higher intensity of chemical weathering. I think this application is on the basis that detrital clay minerals and rock debris in sediments are reliable archives for terrestrial inputs. However, all the K/Al ratios of the Xiamaling Formation reported in this study are higher than that of the UCC (~0.3), seemingly indicating that there was seawater-derived K in these siliciclastic rocks. Please clarify this clearly (also see Major comment 1).

Response: Please see the revised Supplementary Information where we have now included a more detailed introduction to the K/Al proxy. Firstly, lower (rather than higher) K/Al ratios in sediments generally indicate higher intensity of chemical weathering, because of enhanced solubility of K as weathering intensity increases. There are then a number of possibilities for why our K/Al ratios in these two sections are higher than the UCC, which do not necessarily rely on the addition of seawater-derived K. Potassium is commonly enriched in illites (Boyer, *Geophys. Res.*, 1983), and as compiled by Warr (2022), these K-rich clay minerals are preferentially concentrated in mudstones, relative to other coarser-grained sediments. The average K/Al ratio of the UCC is estimated from a compilation of major sedimentary rock types from various tectonic and sedimentological settings (McLennan, *Geochem. Geophys. Geosys.*, 2001). Therefore, relatively elevated K/Al ratios in our two sections, which are dominated by shales and mudstones, may simply be because of the preferential concentration of K in clays. In addition, it is possible that the source sediments to the basin were K-rich in the first place. Because of these possible variables, we use the relative variability in K/Al ratios, instead of absolute values, to indicate relative changes in chemical weathering intensity. This then raises an important observation about the possibility of additional K supply to the sediments from seawater. While this cannot be entirely ruled out (and hence we now mention this in the SI), it is difficult to envisage how such an addition would result in the cyclical trends we see in K/Al ratios

(which is what our interpretation is based on), particularly since these trends coincide so well with other geochemical trends that could not have been affected by this process. Changes in chemical weathering intensity thus provide the most compelling explanation for the K/Al trends we observe.

3. Tables S5 and S6 only present raw geochemical data. As shown in Fig. 2 and discussed in main text, the calculated values of FeHR/FeT, Fepy/FeHR, UEF, MoEF, ReEF, MnEF, P/Al, Preac/Al, TOC/Porg, TOC/Preac, K/Al, and Ti/Al are better to be presented in tables.

Response: Added as requested.

Minor comments:

Line 21–26. I would suggest to introduce the reconstruction result from the lower section first, to the upper section then.

Response: This is a good point and we have changed the order of the two sections throughout the main text in the revised submission.

Line 41. What does “similarly” mean?

Response: We see the confusion here, and have changed this to ‘In addition...’

Line 69. I suggest that at least briefly introduce the background information of the studied area, stratigraphy, and samples, before the geochemical results in the main text, although details have been presented in Supplementary Information.

Response: A brief introduction to the geologic setting has now been added to the main text, as requested.

Line 70–71. As the authors only investigate two short intervals of the Xiamaling Formation, 16 cm of Unit 1 (Section A) and 70 cm of Unit 4 (Section B), thus the words “through unit 1 (Section A) and unit 4 (Section B)” is not appropriate here.

Response: The word “through” has been changed to “from”.

Line 76. As enrichment factors are calculated relative to average UCC, thus “relative to upper continental crust (UCC)” is not needed and should be deleted here.

Response: Deleted as suggested.

Line 80. “values” cannot be “heavier”.

Response: Corrected.

Line 81. How about Re and Mn enrichment factors of Section A?

Response: As shown in the Supplementary Table S5, ReEF ratios are significantly elevated (75 ± 14) throughout Section A, while Mn concentrations are consistently depleted relative to the UCC average, supporting our redox interpretations. Given that neither Re or Mn in Section A provide key additional information (i.e., on top of the existing data we present) for interpreting redox conditions through this section, we prefer not to present this information in the main text to avoid unnecessary length.

Line 88. Fepy/FeHR ratios and MoEF values of Section B are extremely low, which should be also

mentioned here.

Response: Added as advised.

Line 106. Add “those” between “than” and “in”.

Response: Added as advised.

Line 112. “ratios” and “values” cannot be “enrichments”.

Response: We have changed this to “elevated”.

Line 119. You just stated that a sulfidic water column is supported by persistently elevated MoEF values requiring significant sulfide availability in Line 115-116, which seems inconsistent with the sentence here. Please clarify this inconsistency.

Response: The “significant sulfide availability” in Line 115-116 indicates that sulfide abundance during the deposition of Section A was high enough to maintain sulfidic environments where pyrite and thiomolybdate could accumulate. While the phrase “relatively low sulfide availability” in Line 119 means that ocean sulfide availability in the Mesoproterozoic was relatively low compared to euxinic conditions in modern oceans. To improve the clarity here we have amended the sentence to: “These results may relate to relatively low sulfide concentrations compared to modern euxinic basins, which would be an expectation given the low sulfate concentrations proposed for the Mesoproterozoic ocean”.

Line 126. “higher” rather than “heavier”.

Response: Corrected as advised.

Line 128. It is better to present the contrasting reconstruction results of Fe_{HR}/Fe_T ratios and U_{EF} values first.

Response: The contrasting Fe_{HR}/Fe_T and U_{EF} results of Section B are already presented in the Results section.

Line 131–134. I think the high TOC contents in the green mudstone intervals can also be an important piece of evidence for anoxic bottom water conditions.

Response: This is a good suggestion, but we note that although TOC concentrations in the green mudstone intervals are higher than those in the red mudstone intervals, these values are still low (0.08±0.01 wt%). So they are not solid evidence for anoxia in isolation, but do support our interpretations. Overall, we feel it is probably more prudent not to over-emphasize this line of evidence here.

Line 143–144. A fully oxygenated deeper ocean was proposed based on the geochemical evidence of the Unit 3 (Zhang et al., 2016).

Response: Yes, the fully oxygenated deeper ocean model was first suggested by Zhang et al. (2016), based on the geochemical characteristics of unit 3. Then, Wang et al. (2017) also suggested an oxygenated bottom water during deposition of unit 3, and we have cited these papers in the revised manuscript. Our new geochemical data that includes Re (for the first time in terms of application to this unit) suggests dysoxic conditions, rather than fully oxygenated conditions.

Line 160–161. The P/Al ratios of Section A are not low, inconsistent with the supposed P-limited conditions.

Response: We agree, and later in the text we comment on this. The “P-limited conditions sometimes inferred for the Mesoproterozoic” refers to the study of Reinhard et al. (2017), which predicts a TOC/P_{org} ratio of ~300:1 during this period, on a global scale. Here, we are simply compared our P phase association data to the Redfield ratio of 106 and also the predicted value of ~300) to demonstrate that P recycling was promoted under euxinic conditions in the Mesoproterozoic ocean, in the region of study. Actual P/Al ratios at a particular site would reflect the ratio in the local source material, in addition to P drawn down in association with organic matter and/or Fe oxides. As such, a productive, TOC-rich setting might be expected to be characterized by relatively high P/Al, but this does not mean that P was not the ultimate limiting nutrient on the timescale that our sections encapsulate.

Line 164. $\delta^{34}\text{S}$ values can also be included.

Response: Added as suggested.

Line 171. (1) “lower than” rather than “below”. (2) the Preac includes P_{org}, P_{Fe}, and P_{auth}, and thus TOC/Preac ratios should be usually lower than TOC/P_{org} ratios. Why not directly present the P_{Fe} and P_{auth} enrichments instead?

Response: (1) We have changed to ‘lower than’ as suggested. (2) For most settings, P_{Fe} is generally a relatively low contributor to Preac, and P_{auth} is derived from both P_{org} and P_{Fe}, but the degree of sink switching between P phases will vary for each sample. Rather than this degree of detail, what we first need to know to evaluate whether there has been P recycling back to the water column is the overall concentration of the reactive P pool, as this takes into consideration P transfer between different phases during diagenesis. Then, it is not this actual concentration that specifically allows us to investigate recycling, but rather the TOC/Preac ratio, which is then compared to both the Redfield ratio and the suggested ratio of 300:1 for the Mesoproterozoic ocean. In this regard, while TOC/Preac will generally be lower than TOC/P_{org}, this is not really the important point – the important point is the relationship between this ratio and the Redfield ratio. For this reason, P phase partitioning studies always use the approach we have adopted here, since actual concentrations are potentially controlled by many processes.

Line 180–182. Reorganize this sentence.

Response: Rephrased as “Despite this P recycling, sediments in Section A are characterized by persistently elevated P/Al ratios and consistently high TOC concentrations (Fig. 2A), which suggests an overall productive setting throughout deposition.”

Line 190. See the comments on Line 171.

Response: This has been dealt with in our response above.

Line 206. See Major comment 2.

Response: This has been dealt with in our response above.

Line 207. Add “of” between “intensity” and “chemical”.

Response: Added as suggested.

Line 213. “lower” rather than “lighter”.

Response: Corrected as advised.

Line 216–219. Reorganize this sentence. The lower sulfate input would have promoted less intense euxinia, which ultimately lower the flux of recycled P from sediments.

Response: This section has been rewritten for clarity:

“By contrast, periods of less intense chemical weathering would have resulted in a lower influx of both phosphate and sulfate. Whilst a lower sulfate influx would have promoted less intense euxinia, the overall flux of recycled P from sediments remained similar. Therefore, lower phosphate availability in surface waters, largely as a consequence of the lower weathering influx, would have led to relatively lower rates of productivity and TOC burial (Fig. 2A).”

Line 220. See Major comment 2.

Response: This has been dealt with in our response above.

Line 227. Cite Wang et al. (2017) here.

Response: Added as suggested on line 229.

Line 228. Is there evidence for the changes in the upwelling intensity?

Response: There is currently no direct evidence for changes in upwelling intensity, and hence we use the word “possible” here.

Line 232–233. How do you know / What is the evidence, for the two sections in this study being “the deepest part of the Xiamaling basin”?

Response: Sediments from both sections are dominated by very fine-grained rocks, and paleogeographic reconstruction of the NCC indicates that the Xihuayuan area in this study was among the deepest areas during the deposition of the Xiamaling Formation (Qiao and Wang, *Acta Geol. Sin.*, 2014; Wang et al., *Geochim. Cosmochim. Acta*, 2020).

Line 238. Add “Ga” between “1.44” and “to”

Response: Done.

Line 240–244. Unit 4 of Section B was deposited before Unit 1 of Section A, so I suggest to reorganize two sentence.

Response: Reorganized as suggested.

Line 267. “the mixed layer” of what, chemocline?

Response: Yes, but it was not necessary to say this here and we have deleted this text.

Line 273. Need to change to “approaches demonstrate”.

Response: Done.

Line 273–276. Merge two sentences.

Response: Done and changed to: “Our combined approach demonstrates how ocean redox and nutrient cycling varied as a function of location and specific positioning relative to the ITCZ and global atmospheric circulation cells, which drove large-scale variability in ocean redox conditions and associated nutrient cycling, and more subtle, regional-scale orbital timescale variability.”

Line 276. “shorter-”.

Response: Modified as advised.

Line 315–316. Move this sentence before the sentence of “Laboratory standard SWS-3A (BaSO_4 , $\delta^{34}\text{S} = +20.3\%$), inter-laboratory standard CP-1 (chalcopyrite, $\delta^{34}\text{S} = -4.56\%$), and international reference standard IAEA S-3 ($\delta^{34}\text{S} = -32.06\%$) were used to check precision and accuracy”, since you had reported $\delta^{34}\text{S}$ values before.

Response: Reorganized as advised.

Line 326. “RSDs”.

Response: Corrected.

Line 533. Remove “ $\text{K}/\text{Al} = 0.3$ ”.

Response: We have modified this text to note that the UCC K/Al ratio falls below the scale used on our plots.

Figure 2. (1) I suggest to make items of the geochemical profiles of Sections A and B consistent, namely, add ReEF and MnEF profiles for Section A and add MoEF profile for Section B; (2) the line representing the K/Al ratio of the UCC has not been shown and needs to be added.

Response: (1) See our response above. Considering the limited space and that our redox interpretations are robust without the inclusion of these data in the main text, we prefer not to add these additional geochemical profiles; (2) The K/Al ratios are above the UCC average, and we would not be able to visualize the important points on this figure if we alter the scale. Therefore we prefer not to amend this figure.

Figure 4. Marker Shelf 1 and 2 in (B).

Response: New information has been added in the Fig. 4 legend.

Comments on Supplementary Information:

Line 601. “High precision”.

Response: Corrected.

Line 607–608. The Xiamaling Formation is divided into four members, Member 1 to 4 for bottom to top, or into six units, Unit 1 to 6 from top to bottom.

Response: Yes. In this study, we follow the approach of Wang et al. (2017), which divides the Xiamaling Formation into 6 units. In the original manuscript on Line 608, we used “unit 4” because only the first 4 units are mentioned in this study, but we have changed this to unit 6 in the revised

version, to be consistent with Wang et al. (2017).

Line 610. “siliciclastic rocks”.

Response: Modified as suggested.

Line 612–614. The Xiamaling Formation is more than 400 m in thickness. You should explain why you only focused on two short intervals of Unit 1 (Section A) and Unit 2 (Section B).

Response: A previous study has identified longer-term orbital forcing in unit 3, utilizing relatively simple total elemental analyses (Zhang et al., Proc. Natl Acad. Sci. USA, 2015), while Wang et al. (2017) performed more detailed redox analyses, but at much lower resolution. In fact, orbital timescale cyclicality has not previously been identified in unit 4 (our Section B). Our aim with this study was to explore periodicity in sediments (other than those from unit 3) at very high resolution, but with a specific focus on interactions between ocean redox and nutrient cycling, representing two key perspectives for studying early Earth evolution. These are much more complicated and time-consuming analyses, and since our aim was to examine links between these factors, in addition to their drivers and potential implications, it was only necessary (and practically possible, given the complexity of the geochemical methods) to focus on one or two individual cycles.

As suggested, we have added additional text to lines 647-654 to highlight this: “Our focus on these relatively small intervals allows us to perform multiple, very high-resolution geochemical analyses across two complete cycles in each case. This approach builds upon reported evidence for orbital cyclicality through unit 3 of the Xiamaling Formation, based largely on major element data (Zhang et al., Proc. Natl Acad. Sci. USA, 2015), and on lower resolution analyses of redox conditions through the Xiamaling Formation (Wang et al., Am. J. Sci., 2017). Thus, our approach, which additionally incorporates novel biogeochemical modelling in relation to palaeogeography and climate forcing, provides the first detailed evaluation of both the controls on apparent redox cyclicality across individual cycles, and implications for nutrient cycling.”

Figure S1. Please add plotting scales for Sections A and B.

Response: We have the pencil (~15 cm) in Fig. S1A and the ruler (30 cm) in Fig. S1B as scales for sections A and B, respectively. However, as requested we have added a scale bar to both sections.

Reviewer #3 (Remarks to the Author):

General comments:

Song et al. present a variety of high resolution data exploring the influence of climate fluctuations on geochemical cycling within a Mesoproterozoic environment. Data include a variety of analyses (Fe spec, P spec, S isotopes, major and trace elements, and TOC) and are of good quality. Interpretations of redox state are very solid; authors do an excellent job of integrating findings from each analysis into a coherent geochemical story. The authors propose orbital scale climate variation in the form of ITCZ and Hadley cell changes as drivers of redox and nutrient fluctuation in their study environment. These appear to be reasonable hypotheses. The article's supplementary information is very useful and goes into great detail about aspects of the paper. Additionally, the manuscript is very well written and pleasant to read.

Response: We are very grateful for this highly supportive appraisal of our work and its significance.

We did notice that the methods, results, and some interpretations in this manuscript are similar to those in Wang et al. 2017. We recognize that this manuscript brings new insights, so we just recommend that it be made explicit whether or not any of your data derive from Wang et al. 2017 and how your paper builds upon this previous study.

Response: This is a good point and we are very happy to have the opportunity to further emphasize the novel aspects of this work. None of the geochemical data we report originate from Wang et al. (2017). As we now highlight in the SI (lines 647-654), our approach directly builds upon the very limited work that has been published in relation to orbital cycling in the Mesoproterozoic. We now specifically state that “Our focus on these relatively small intervals allows us to perform multiple, very high-resolution geochemical analyses across two complete cycles in each case. This approach builds upon reported evidence for orbital cyclicity through unit 3 of the Xiamaling Formation, based largely on major element data (Zhang et al., Proc. Natl Acad. Sci. USA, 2015), and on lower resolution analyses of redox conditions through the Xiamaling Formation (Wang et al., Am. J. Sci., 2017). Thus, our approach, which additionally incorporates novel biogeochemical modelling in relation to palaeogeography and climate forcing, provides the first detailed evaluation of both the controls on apparent redox cyclicity across individual cycles, and implications for nutrient cycling.”

In addition, we initially mention the limitations of previous work on lines 54-57 of the revised text. We also note here that while our nutrient P phase association analyses have never been performed at this resolution through any Mesoproterozoic succession (and to our knowledge have only been applied to a few samples from the 1.1 Ga Taoudeni Basin; Guilbaud et al., Nat. Geosci., 2020), and hence are entirely novel, many of our conclusions on cyclicity and the nature of the redox conditions through the Xiamaling Formation are also somewhat different to previous reports. For example, the previous work of Wang et al. (2017) has a very different interpretation of unit 4 that does not involve orbital cyclicity, and additionally we propose a revised interpretation for the redox conditions in terms of the development of dysoxic, rather than oxic, deep water conditions. Thus, our study suggests that orbital fluctuations in redox conditions were a fundamental feature of the Mesoproterozoic ocean, rather than being restricted to certain high productivity settings.

In sum, there have been no studies that systematically unravel links between orbitally-driven controls on redox cyclicity and associated nutrient cycling through the Mesoproterozoic (or indeed, the entire Precambrian). The addition of biogeochemical modelling also takes our study to a new level, by demonstrating in a quantifiable manner, how changes in the position of the ITCZ could have driven the observed changes. We also note that in contrast to previous studies, we also make a very strong case that our observed changes speak directly to the apparent heterogeneous nature of ocean redox reconstructions through the Mesoproterozoic (and indeed, the Precambrian in general). While this might seem obvious, it is clearly not a factor that has generally been considered in the many published studies on the redox evolution of the ocean and atmosphere through this time period. Thus, we feel that our study is a major step forward in many regards, and we hope that the revised manuscript adequately portrays this.

All in all, we believe this manuscript is of high quality and worthy of publication. We propose some minor changes. In a few places, we suggest the authors should qualify some of their hypotheses and/or acknowledge alternative scenarios. We also suggest they include some additional information in places, but nothing critical to the paper's main points.

Response: Many thanks again, and we hope we have addressed all of the suggestions in a satisfactory manner.

Sincerely,

Amy Hagen and Benjamin Gill

-

Specific comments (also included in attached Word doc):

Line 118-123: We are not sure you can differentiate their proposed scenario that invokes local changes in sulfide availability driving Mo enrichments from one where the enhanced weathering also just delivers more Mo to the basin (with the sulfate). Both would lead to intervals with higher Fe(pyrite)/Fe_{HR} ratios, lower $\delta^{34}\text{S}$ of the pyrite and higher Mo enrichments. We would recommend the authors also discuss this possibility; it doesn't greatly change their interpretations.

Response: This is a good point that has encouraged us to think more deeply about this aspect of the data. In fact, based on similar high-resolution data applied to euxinic Cretaceous OAE2 sediments (Poulton et al., *Geology*, 2015; Krewer et al., *Am. J. Sci.*, in review) we do think we can discount the possibility the reviewers allude to in this comment. More intense chemical weathering (lower K/Al) appears to have occurred during what we have termed the more highly euxinic intervals. Higher intensity chemical weathering increases the oceanic influx of both sulfate and Fe_{HR} (in the form of Fe oxides) from pyrite weathering on land, but Fe_{HR} is proportionately increased relative to sulfate as weathering intensity increases, due to additional release of Fe_{HR} from parent silicate minerals (Canfield, *Am. J. Sci.*, 1996; Poulton and Raiswell, *Am. J. Sci.*, 2002; Poulton and Canfield, *Elements*, 2011; Poulton et al., *Geology*, 2015). If sulfide concentrations in the water column remained constant, this would increasingly focus the locus of Fe_{HR} sulfidation to deeper in the sediment pile, with proportionately less sulfidation (i.e., pyrite formation) occurring in the water column. If this were the sole process at play, it would, in turn, result in increased $\delta^{34}\text{S}_{\text{py}}$ values as proportionately more of the pyrite would be formed under closed system conditions in the sediment. But this is opposite to what we see.

Our data require proportionately more pyrite to have been formed under well mixed, open system conditions in the water column, giving the observed lighter $\delta^{34}\text{S}_{\text{py}}$ values. This occurs despite the likelihood that if sulfide concentrations had remained constant in the water column, higher $\delta^{34}\text{S}_{\text{py}}$ values would be expected, as discussed above. Therefore, this implies that sulfide concentrations were significantly higher during our 'highly euxinic' intervals because it is sulfide concentration that directly influences the rate of Fe_{HR} sulfidation (e.g., Canfield et al., *Geology*, 1992; Poulton et al., *Geochim. Cosmochim. Acta*, 2004). Thus, higher concentrations of water column sulfide were required to focus more of the Fe_{HR} sulfidation in the water column, rather than the sediments. While the reviewers are correct that Mo fluxes could also have been increased due to enhanced weathering, our revised analysis specifically suggests that sulfide concentrations would have been a major driver of the fluctuations in Mo_{EF} values. This is a rather complicated and lengthy explanation, and so we have decided to amend the main text slightly, and explain this in more detail in the SI, with reference to this in the main text.

Line 126-127: It might be nice to show a cross plot or stratigraphic plot with only these two sets of data in the supplement to more clearly illustrate this point.

Response: A cross plot of Mo_{EF} vs $\delta^{34}\text{S}$ does not show a clear relationship, possibly because these

parameters are slightly offset in terms of the timing of their response to biogeochemical perturbations. However, their general stratigraphic relationship is shown quite clearly on Fig. 2A.

Line 129: It might be best to qualify what is meant by “mineralogy” here, since Fe spec targets operationally defined pools and therefore we don’t know specific minerals.

Response: We completely agree and should not have used the term ‘mineralogy’ in this context here. We have rephrased this part of the text, and have also altered the Methods to highlight that the technique ‘targets’ Fe phases, rather than quantitatively extracting certain minerals in each step.

Line 131-134: Given the higher Fe/Al ratios (of the green intervals versus those in Section A), it might be worth mentioning the potential for clay to be a sink of highly reactive Fe.

Response: We don’t feel we have the additional data to address this here and so prefer not to speculate on this.

Line 175-178: I’m not sure ‘rate’ is the correct word here. To compare rates we need to know the amount of time it took to produce the TOC/P ratio. ‘Amount’ might be a better word.

Response: Changed to ‘a lower extent’.

Line 179-180: This is probably not a realistic assumption. It’s fine that it was assumed (it may be the best we can do), but we would like to see this statement better qualified.

Response: We have amended the main text to ‘relatively constant sedimentation rate’, and now expand on the limitations of this in the SI. Overall, there is no evidence for a change in grain size across this 16 cm interval, and hence it is unlikely that any change in sedimentation rate would have been sufficient to alter our conclusions.

Line 209-210: It would be better to cite the study/studies which ground truth Ti as a dust proxy rather than these two sources which simply apply the proxy.

Response: Agreed. However, the use of Ti/Al ratios is quite deeply embedded in the literature, and these ratios are commonly used in aeolian source studies, with its potential commonly stated alongside direct application. The Yarincik paper we cite is a particularly commonly cited paper in this regard. However, we have added an additional reference here that is more fundamental in nature.

Line 217-218: Shouldn’t P recycling have declined as organic matter production declined?

Response: P recycling does not have to be directly related to organic matter production. The main control is specifically the generation of sulfide, particularly close to the sediment-water interface. In our sediments, there appears to have been a balance whereby the actual flux of recycled P was approximately similar during weakly and more strongly euxinic conditions, possibly because under strongly euxinic conditions more sulfate was utilized in the water column and hence there was less remaining to fuel sulfate reduction in the porewaters, but this is speculative.

Line 220-221: Low chemical weathering rates seem at odds with the elevated Fe/Al ratios for the green intervals in Section B compared to all of Section A. May suggest that either there is a difference in the weathering flux of iron (source or intensity) between these sections.

Response: Total Fe concentrations in the green intervals in Section B are very similar to those of

Section A (Supplementary Table S5). The relatively elevated Fe/Al ratios in green intervals occur largely due to lower Al concentrations in Section B. Al is widely used as a terrestrial input proxy, and this supports our interpretation that weathering intensity in Section B was lower than in Section A.

Line 224: The authors should be specific here. What large-scale changes are they referring to: overall weathering rates, type of material being weathered, and/or amount of geographic area being weathered?

Response: Here we were referring to a change in the precise geographic area being weathered, and can see why the sentence was a little confusing. We have rewritten for clarity: “Indeed, since there is no apparent reason to assume large-scale, repetitive changes in the geographic area being weathered (i.e., the sediment source area) during deposition of Section B, peaks in Ti/Al around the mid-point of green mudstone intervals likely document maxima in the aeolian source flux.”

Line 232-234: I’m not sure you can rule this out. Estimating paleowater depths is difficult, so I don’t think you can simply say it is not a factor. Changes in terrigenous input could be a factor.

Response: The red/green mudstones of Section B and black shales of Section A are very fine-grained sediments, and are considered to have been deposited in deep waters. It is certainly possible (and likely) that sea level fluctuated, and as the reviewers note, estimating paleowater depth with any degree of accuracy is notoriously difficult. However, as clarified in the Supplementary Information, both sections appear to have been deposited well below storm wave base, and therefore, sea level changes are unlikely to have been a significant factor controlling the geochemical differences between two sections. We agree that variations in terrigenous input could drive the distinct contrasts in ocean biogeochemistry between the two sections, and we discuss this in the “Orbital forcing of climate change” section. To address the first point, we have amended the text to be more explicit: “and hence a sea-level control on the biogeochemical dynamics is highly unlikely”. This leaves the possibility that there was sea level change, but highlights that our sections from the deepest part of the basin were unlikely to have been greatly affected by any such change.

Line 235-236: This seems to be a reasonable hypothesis, but are there potentially other climate influence factors? It may be worth mentioning some alternatives.

Response: See response to the last comment. Sea-level change could be a factor, but appears unlikely as mentioned in the text. We are unclear what other hypotheses might be put forward to explain the data. The hypothesis we propose appears the most likely explanation for our data.

Line 250: It could be useful to mention why you’ve chosen to couple these two variables or what model outputs would look like if you changed just one of these factors.

Response: This is a good point. In this simple model all deep water upwells through the shelf boxes (with no open ocean upwelling), so the connection to the deep ocean is likely overly strong. This acts to buffer against any changes driven by weathering inputs. Thus, to explore a high and low P scenario on the shelves we decided to limit upwelling to the low P shelf. We now state this and show a model version in the SI where we only change the weathering rates. This still agrees with our key conclusions on the redox states of these margins, but the strong connection to the deep ocean P supply means that the oligotrophic model shelf displays oscillations on the same timescale as the eutrophic box.

Line 278-280: This seems like a bit of an overgeneralization, I'd suggest changing the wording a little bit here.

Response: We feel this is a very important statement to make with regard to this work, and it is certainly something that has generally not been factored into studies of Mesoproterozoic redox heterogeneity. We specifically state that “climate forcing has *rarely* been factored into studies”, and “which instead *often* assume”, and feel this gives sufficient leeway such that the statement is not an over-generalization.

Line 329: I am not a box model expert so I can't offer much feedback here, but all looks good and seems to be explained well.

Response: We appreciate the supportive comment.

Line 521: I would suggest using something other than just a red arrow to indicate the studied sections. Perhaps a curly bracket would be helpful in showing that you're zooming in on a portion of the big column?

Response: Modified as suggested.

Line 572: I'd suggest adding a reference here so we know why you've shown that particular range for the ITCZ.

Response: We have added a sentence and two references: “The ITCZ generally falls within $\sim 15^\circ$ of the equator in both hemispheres^{59,60}”.

Line 678: Again, I'm not sure this is likely to be true. After all, you just discussed turbidites which represent episodic sedimentation. It could be good to just put in a short note acknowledging this.

Response: We agree that sedimentation rates will have varied, but we think at this point it is useful to give a very broad approximation of the magnitude of time the sediments encompass. Given likely changes in sedimentation rate, we have specifically taken the step of not assigning a particular orbital frequency to our intervals of study, as discussed in more detail in this section.

Line 734-735: Since you mention this, it might be useful to elaborate a bit on why you don't see this being a problem in your section.

Response: Good point. We have added: “which we note are very minor in our samples and hence would not significantly affect the background geochemical data; Fig. S1”.

Line 738-739: This section might be a bit confusing for those who haven't read Pasquier et al. I'd suggest adding a bit about what they claim in the paper before you proceed to counter it. However, I also think that the vast majority of papers support Fe spec as a proxy and you may not need to dedicate so much space to addressing this. I think your last 2 sentences sum up your point nicely. This is just an opinion and not critical to your paper so feel free to keep this section as is if you like.

Response: We appreciate the balanced suggestions here, and are loath to give space to the Pasquier et al. publication since it is so fundamentally flawed. However, until the community response to this paper (which is currently being prepared) has been published, we feel it is important (if frustratingly so) to present text like this, for the readers who are not experts on Fe speciation, but may have seen the Pasquier et al. publication. We have taken on board the suggestion here, and the text has been

amended to give more context: “A recent study has argued that Fe speciation data may be compromised by diagenetic processes²⁵”.

Line 742-746: As mentioned above, I think these two sentences emphasize an important point: that regardless of any doubts in Fe spec, your other data fully support your interpretations.

Response: We agree, and are grateful for the confirmation.